# Validation of Pelvis and Trunk Range of Motion as Assessed Using Inertial Measurement Units

**DOI:** 10.3390/bioengineering11070659

**Published:** 2024-06-28

**Authors:** Farwa Ali, Cecilia A. Hogen, Emily J. Miller, Kenton R. Kaufman

**Affiliations:** 1Department of Neurology, Mayo Clinic, Rochester, MN 55905, USA; 2Department of Orthopedic Surgery, Mayo Clinic, Rochester, MN 55905, USA; hogen.cecilia@mayo.edu (C.A.H.); miller.emily@mayo.edu (E.J.M.); kaufman.kenton@mayo.edu (K.R.K.)

**Keywords:** inertial measurement unit, validity, reliability, motion capture, validation study, reproducibility of results

## Abstract

Trunk and pelvis range of motion (ROM) is essential to perform activities of daily living. The ROM may become limited with aging or with neuromusculoskeletal disorders. Inertial measurement units (IMU) with out-of-the box software solutions are increasingly being used to assess motion. We hypothesize that the accuracy (validity) and reliability (consistency) of the trunk and pelvis ROM during steady-state gait in normal individuals as measured using the Opal APDM 6 sensor IMU system and calculated using Mobility Lab version 4 software will be comparable to a gold-standard optoelectric motion capture system. Thirteen healthy young adults participated in the study. Trunk ROM, measured using the IMU was within 5–7 degrees of the motion capture system for all three planes and within 10 degrees for pelvis ROM. We also used a triad of markers mounted on the sternum and sacrum IMU for a head-to-head comparison of trunk and pelvis ROM. The IMU measurements were within 5–10 degrees of the triad. A greater variability of ROM measurements was seen for the pelvis in the transverse plane. IMUs and their custom software provide a valid and reliable measurement for trunk and pelvis ROM in normal individuals, and important considerations for future applications are discussed.

## 1. Introduction

An adequate trunk and pelvis range of motion is essential for healthy independent living. Limited mobility, or a restricted range of motion, can occur due to musculoskeletal or neurological disorders that, together, form the largest burden of disability-adjusted life years lost [1,2]. Mobility limitations contribute to disability for many reasons. Individuals with poor mobility experience a loss of independence due to the need for assistance in activities of daily living [3,4]. Inefficient biomechanics in individuals with limited mobility can lead to abnormal biomechanics, resulting in the expenditure of more energy, causing fatigue [5]. Mobility impairments can contribute to cognitive burden in those with neurodegenerative disorders, compounding their disability [6,7]. These mechanisms alone or in combination add to the risk of falls, and the number of fatal falls among individuals over the age of 65 years has doubled since the 1990s [8]. Adequate trunk and spino-pelvic ROM is essential for activities of daily living, gait, and stability [9,10,11]. In the aging individual, there may be an almost 50% reduction in spinal range of motion (ROM), leading to restricted mobility and abnormal gait and balance [12]. Adequate mobility also correlates with function and quality of life in neurological diseases such as Parkinson’s disease [13]. Therefore, measurement of ROM is important for assessing functional capacity.

Motion capture (mocap) systems are commonly used to assess ROM; however, inertial measurement units (IMUs) are becoming more popular. The IMU is an attractive option due to its ease of use, lower cost, and no need for highly trained personnel or a large amount of space. While IMU systems have been evaluated in a variety of use cases and environmental settings (laboratory, clinic, or home), most prior studies have used a custom IMU configuration and custom analysis pipelines that are only possible with biomechanical expertise, thereby limiting generalizability [14,15,16]. For example, our lab previously compared the IMU using a custom code and found the static sensor orientation to be within 0.6 degrees of gold-standard custom apparatus, and a precision of 0.1 degrees, whereas angular velocity was accurate to within 4.4 degrees per second and 0.2 degrees precision [17]. Studies replicating a real-world clinical office visit, where biomechanical expertise may not be available and a commercially available IMU system and software would be utilized, are limited. In laboratory settings, IMUs have been found to be valid and reliable for upper body kinematics of trunk, shoulder, and elbow [18,19,20]. Lumbar kinematics measured using an IMU have also been found to be reliable between test and retest [21,22]. In a study, Franco et al. used a series of five spinal IMUs and found the RMSE of sagittal ROM to be 2.1 degrees and frontal ROM to be 2.3 degrees in 11 participants [23]. Among neurological diseases, IMU systems have been shown to capture disease progression in Parkinson’s disease, and measurements of ROM correlate with axial rigidity and overall function [13,24]. In patients with ataxia, IMUs were compared to pressure sensitive walkways, and measurements captured the variability in gait of ataxic patients beyond the degree of measurement error between devices. Hence, IMUs can detect clinically meaningful change in kinematic and spatial–temporal variables [25]. However, the accuracy and reliability of ROM measurements using a commercially available IMU system and its software application in the absence of custom calibration and time series data manipulation is unexplored.

Existing validation studies have used custom IMU set ups, variable number and location of sensors, and custom analysis algorithm, limiting the generalizability of the findings. Our study is novel and important as we assess the accuracy and reliability of kinematic measurements performed using an IMU system and its companion software (Mobility Lab version 4). As the clinical and research use of IMUs increases, clinicians who do not have access to a biomechanical lab, custom IMU sets ups, and analytics pipelines are highly likely to rely on commercially vended software solutions that accompany IMUs to calculate kinematics. An assessment of how such software perform in normal individuals is the first step before application in individual disease groups can be undertaken. In this study, we evaluate the accuracy (validity) and reliability (consistency) of the trunk and pelvis ROM in all three planes (frontal, sagittal, and transverse) during steady-state gait (self-selected walking speed without active acceleration or deceleration), as measured using an Opal APDM 6 sensor IMU system and calculated using Mobility Lab version 4 software, compared to a gold-standard optoelectric motion capture system. Another novel aspect of this study is the use of a triad of markers placed directly on the IMU (sternum and sacrum) for a direct comparison of ROM calculated using the triad trajectory to the IMU. Finally, prior studies seeking to validate Mobility Lab only report correlations [26]. However, we report the magnitude of error, agreement, and reliability to provide a more thorough evaluation of sensor performance. This study also provides a validation protocol that can be used in future studies to assess the performance of an IMU system in different normal and pathological gait assessments.

## 2. Materials and Methods

A random convenience sample of 13 healthy participants (7 female, mean age 32 (SD 9.6) years, BMI: mean 26.5 (SD 4.5)) was recruited in this study. Since our goal is to specifically assess the validity and reliability of the custom software accompanying the IMU system, we included healthy adults, older than 18 years of age with normal gait. We excluded individuals who may have a neurological- or orthopedic disease-related gait abnormality, or advanced aging-related gait decline, which would introduce a variety of confounders when evaluating the output of Mobility Lab. Participants provided written, informed consent and the study was approved by the Mayo Clinic institutional ethics review board. This sample size is consistent with other studies reporting the validation of IMUs that usually range from 5 to 25 subjects.

The APDM Opal IMU (APDM Wearable Technologies, Portland, OR, USA) was used in this study. The subject donned six IMUs (sternum, sacrum, bilateral dorsum of feet, and bilateral wrist) oriented according to the company’s recommendations (Figure 1). This configuration is required for the Mobility Lab software to compute gait data. To minimize the relative movement between each sensor and the subjects, double-sided tape was used to secure the IMUs directly to the skin, except for the wrist IMUs, which were secured with the provided straps. APDM’s commercial data collection and processing software, Mobility Lab (version 4), generated discrete measurements of ROM using a proprietary algorithm in the sagittal, coronal, and transverse planes. The Mobility lab software uses a proprietary algorithm to generate the motion matrix based on the data from all six sensors.

Retroreflective markers (Figure 1) were placed on the subjects for kinematic analysis using mocap. The specific location of these anatomical markers followed the landmarks identified using the Helen Hayes configuration originally described by Davis et al. [27]. Additionally, a triad of retroreflective markers (Figure 2) was placed directly on the sternum and sacrum IMUs for a subset of 5 participants for a 1:1 comparison of the IMU trajectory. Kinematics were captured at 120 Hz with a 14-camera optical mocap system (Raptor-12, Motion Analysis Corporation, Santa Rosa, CA, USA). Local coordinate systems for the anatomical trunk and pelvis segments were defined from the static pose and were tracked dynamically using commercial software (Visual3D, v2023.10.1, C-Motion, Inc., Germantown, MD, USA). The frontal (*y*-*z*) plane of the anatomical trunk segment for motion capture was defined as the best-fit of the bilateral acromion and bilateral anterior superior iliac spine (ASIS) 3D markers in the static pose. The local segment coordinate system was then defined with an origin at the midpoint of the acromion markers; the inferior–superior (+*z*) axis lay on this plane, in the direction of the mid-ASIS to segment origin. The *x*-axis was defined as normal to the frontal plane in the anterior direction, and the *y*-axis was the cross product between the *x*- and *z*-axes. The dynamic motion of the anatomical trunk segment was tracked with the bilateral acromion and right scapula markers. The anatomical pelvis segment coordinate system was defined using the Helen Hayes model with 3D markers on the bilateral ASIS and bilateral posterior superior iliac spines (PSIS); with the origin placed at the midpoint of the bilateral ASIS, the *x*-axis oriented anteriorly, the *z*-axis oriented superiorly, and the *y*-axis was defined as the cross product of the *x*- and *z*-axes. The 3D movement of the anatomical pelvis segment during dynamic trials was tracked with the bilateral PSIS and the sacrum markers. The models in mocap to create trunk and pelvis anatomical segments are depicted in Figure 3. This was compared with pelvis ROM measured using the IMU. It should be noted that the IMU software (Mobility Lab version 4) output is labeled “lumbar spine ROM”. However, in our procedures, the IMU location was standardized such that it was placed between the participants’ bilateral PSIS, centered on the sacrum; therefore, the ROM recorded using the IMU in this study is the pelvis, not the lumbar spine, motion to enable a direct comparison with mocap. We offer this clarification to enable the replication of our work by other users of Mobility Lab.

The measurement of a triad coordinate system placed directly on the IMU was performed for a subset (n = 5) of participants (Figure 2). For each marker triad (sternum and sacrum IMU), a local coordinate system was defined from the static pose with the origin at the center marker, on a plane (*y*-*x*) defined by all three markers; where the *x*-axis was oriented anteriorly, the *z*-axis was oriented superiorly, and the *y*-axis was oriented laterally to the participants’ left. The triad segments’ 3D motion was tracked with all three respective triad markers in the dynamic trials.

Kinematic angles for all mocap segments (anatomical and triad, trunk, and pelvis) were calculated as a local coordinate system with global respect to each plane of motion (frontal *x*, sagittal *y*, and transverse *z* planes) for two steady-state strides of each walking trial. Steady-state gait is representative of comfortable walking at a self-selected speed without acceleration or deceleration. Mocap assesses strides in the middle of the recording volume as a way of capturing steady-state. The ROM for each segment, and in each plane, was then taken as the difference between the maximum and minimum segment angle observed during the walk (degrees).

Participants were instructed to walk at a comfortable pace across the 7 m, level walkway when the Mobility Lab software sounded a tone. This was repeated for a minimum of eight walking trials. One walking trial is defined as the patient walking down the entire length of the 7 m laboratory walkway once. For all participants (n = 13), the trunk and pelvis ROM was measured using Mobility Lab and the anatomical mocap model, and over five best steady-state walking trials were selected for further calculations. All recorded trials were assessed for data quality (IMU and mocap data were appropriately time-synced, and the walking pass dataset was complete for both systems). Trials that were incorrectly recorded, or had incomplete data (e.g., incorrect timing of start/stop, or data stopped streaming to either system) were excluded. From the remaining trials, preference was shown towards trials recorded later in the session, under the assumption that those were more representative of the participants’ steady-state walk. For the subset of participants with the triads (n = 5), the trunk and pelvis ROM measured using the triad mocap model was also assessed for these trials. A physical therapist accompanied the patient at all times during the test and verified that the markers and IMUs remained affixed to the correct location at all times. Any marker movement was immediately corrected.

The validity of the trunk and pelvis ROM was compared using Bland–Altman plots to demonstrate the absolute error in the frontal, sagittal, and transverse plane. Limits of agreement (LoA) were also calculated for the two modalities to assess agreement between measurements obtained [28]. To assess reliability, root mean square error (RMSE) was calculated for the IMU and mocap using a method previously described by Camp et al. [29]. Interpretation of the RMSE was conducted according to the Poitras method, where an error of <5 degrees represents an excellent reliability, and an error of <10 degrees represents a good reliability [30].

## 3. Results

The number of trials completed for each participant ranged from 8 to 12 (average 9.7). The range, median, and interquartile range are also shown below in Table 1.

### 3.1. IMU versus Clinical Mocap

The trunk and pelvis ROM measurements obtained via IMU were compared to the mocap for all three planes for 13 participants (Figure 4). The IMU measurements were found to be accurate, with small errors observed. The error values were under 5 degrees for all measurements, except pelvis sagittal and transverse plane ROM, where they were within 10 degrees. There was no systematic error seen across the range of ROM values.

Based on the mean differences shown in Figure 4, all mocap measurements were slightly larger than IMU measurements, except for frontal and sagittal trunk ROM, where IMU values were slightly larger. Limits of agreement were used to compare measurements obtained using the two methods. The agreement was within 5 degrees for all ROM values, with a greater difference seen in pelvis sagittal and transverse ROM values, which were still within 10 degrees. Therefore, we conclude that the IMU measurements were overall valid for pelvis and trunk ROM assessment compared to mocap; however, a greater variability and larger margin of error may be seen in the pelvis sagittal (anterior and posterior tilt) and transverse (rotation) plane ROM.

The reliability or consistency of mocap and IMU were both excellent (Table 2). All values were within 5 degrees for IMU, which seems to overall have a lower variability in measurements. Mocap captured a wider range of values for pelvis sagittal, frontal, and trunk transverse ROM.

### 3.2. IMU versus Triad Mocap

A comparison of the trunk and pelvis ROM as calculated using the triad trajectory compared to the IMU was also made in all three planes for five participants (Figure 5).

IMU measurements were found to be accurate compared to the triad mocap. Differences were within 5–10 degrees; however, a greater variability was seen in all measurements specifically for the transverse plane pelvis ROM. The differences were centered around zero with small mean difference values, showing that slightly higher measurements were obtained using the triad than mocap, but overall errors were small.

A wider range of values was seen for the mean difference between triad- and IMU-acquired ROM values. However, the difference in trunk and pelvis ROM measured using both modalities was small, except for pelvis ROM in the transverse plane, which showed the largest variability in measurements. Since the IMU measurement will be the same, the general trend revealed that more variability was seen in the measurement for the triad, particularly for pelvis range of motion in the sagittal and transverse planes.

The agreement between the two modalities was also high (Figure 5). Due to only five subjects in the triad group, the RMSE was not calculated for this group.

The triad measurements had overall more variability than the clinical anatomical model mocap or IMU. Mocap considers multiple bilateral markers to construct an anatomical segment, hence may smooth out some of the variability. The IMU system had the least variability or spread in the data. The reason for this difference is likely related to the method of computation of the ROM values. The triad mocap measured the trajectory of a set of three markers placed on top of a single IMU to most closely mimic IMU trajectory. However, in contrast to the triad itself, the IMU had a much lower variability, likely due to Mobility Lab’s proprietary motion matrix computation of ROM using data from all six sensors. This suggests that the reliability of measurements using the IMU may be better, but an important limitation is the potential loss of biologically meaningful variation in the data, as well as a lack of reproducibility of a proprietary software methodology.

## 4. Discussion

In this study, we measured the accuracy and reliability of the trunk and pelvis ROM measured using an IMU (APDM Opal 6 sensor suite) and its proprietary software package, Mobility Lab (version 4), compared to mocap. The data demonstrated that an IMU provides valid and reliable measurements for trunk and pelvis ROM. IMUs and their proprietary software systems provide an easy and cost-effective tool for clinical use. However, validating these measurements is essential to ascertain a degree of error. We compared IMU measurements to mocap in two ways. We used a triad of retroreflective markers placed on the IMU for a direct comparison of trajectory. We also used a typical clinical mocap model with a full body set of retroreflective markers, where the anatomical trunk segment and anatomical pelvis segment was defined using the Helen Hayes model to calculate trunk and pelvis ROM. The pelvis ROM had a greater variability overall in all planes. A greater variability was captured using the triad; the least variability was captured using the IMU. The ROM values tended to be larger for the mocap than the IMU values.

The IMU and its proprietary software provided accurate and reliable estimates for trunk and pelvis ROM. This finding is consistent with prior reports where lower body joint kinematics calculated using the IMU were within 5 degrees of motion capture [31]. Shull et al. compared various wearable devices and found that the consistency was high and RMSE was lowest (2–4 degrees) for truncal motion measured using an IMU [32]. Most previous studies report a strong correlation between IMUs and the gold-standard mocap with an intra-class correlation coefficient as high as 0.8 [33,34,35]. A more relevant measure is the accuracy in terms of absolute error or mean difference from the gold-standard. Errors for trunk range have been reported from as low as 0.7 degrees to as high as 4.5 degrees [36,37]. The most common reports agree with our findings with a thoracic trunk ROM of 2.4 to 2.6 degrees [38]. The pelvis tilt and obliquity measured using IMUs have been assessed to be within 5.0 degrees of mocap during walking [39]. Pelvis and trunk ROM values measured using IMU versus mocap also have a high agreement during dynamic tasks [40,41]. However, the accuracy of the measurements may vary in disease groups and during certain transitions such as sit-to-stand [42]. The output of the Mobility lab software was comparable to these previous reports that utilize custom IMU configurations, data filtering, and analysis.

In our study, the pelvis measurements had a wider range of variability compared to the trunk. Variability was more pronounced in the transverse plane than the frontal or sagittal plane for both trunk and pelvis. Similar to our findings, Plamondon et al. found that the error was less than 3 degrees for trunk motion in the frontal and sagittal plane, but was as high as 6 degrees for rotational movements about the transverse plane and depended on the duration of the tasks examined [35]. Our findings are in agreement with another group that found more reliable sagittal and axial plane measurements compared to less reliable rotational movements [43]. There may be many reasons for this finding. Participants may have variable lumbo-pelvic interactions during weight-bearing gait [44]. For example, individuals with lower back pain or variable hip strength demonstrate an altered phasic interaction between the spinal and pelvic segments during gait and other tasks [43,45,46]. Staszkiewicz et al. also demonstrated that the walking surface can impact these measurements [47]. Finally, there may be device-related factors that affect rotational measurements with IMU and mocap technologies [48]. Measurements obtained using the triad were also more variable with a wider spread of data in all planes of motion. We suspect that this may be due to the triad itself, as it is capturing local marker trajectory, whereas, in comparison, the anatomical segment derived using the Helen Hayes system in mocap uses bilateral markers at the pelvis and shoulders to create an anatomical segment from which ROM is calculated. It is also important to note that IMU measurements had the narrowest range of variability in our study. These findings are in concordance with another study that showed a lower variation in kinematic measurements obtained using an IMU [49]. The reduced variability in the IMU may be a result of computational methods used by the Mobility Lab proprietary algorithms, and while this enables reproducibility, there may be a loss of biologically meaningful variability in the data. For example, in diseases with large trunk and pelvis ROM, such as spina bifida [50,51], the limited variability captured using the IMU may lead to the underestimation of the true value, leading to inaccuracies. Moreover, ROM measurements of the pelvis taken using an IMU may vary in different positions such as the flexed seated position, which showed a greater variation [52]. Therefore, validation assessments under walking conditions may not apply to other specific motion analysis tasks or activities. Previous studies also show that IMU accuracy can be affected by body mass index (BMI), and is potentially lower in individuals with a higher BMI [53]. We did not find an effect of BMI or other demographic factors in this study to explain the larger errors, but our sample size was small. Evaluation of anthropometric factors should be included in future larger sample size studies as it may affect IMU performance.

When evaluating IMU performance in a specific disease, device-related errors and variability of measurements should be evaluated in the context of clinical heterogeneity and the magnitude of minimal clinically significant changes. For example, in individuals with Parkinson’s Disease, reduced trunk ROM has been associated with falls, but the average difference between fallers and non-fallers was less than 5 degrees [54,55]. This difference may be clinically significant or truncal ROM may impact overall gait, balance, and stability, leading to the noted difference in the functional state [13]. Similarly, in lumbar spinal stenosis, post operative outcomes varied by group level with a difference in spinal ROM of 5–6 degrees [56]. Igawa et al. also described how other aspects of gait and posture may influence truncal ROM measurement, such as hip extension angle and even step width [57]. In neurodegenerative movement disorders such as Parkinson’s Disease, change in kinematic and spatial–temporal parameters as measured using IMUs have been shown to correlate with clinically meaningful change in neurological function [24,58]. Even though the Opals and Mobility Lab for kinematic assessment appear to be valid in normal individuals, their performance and ability to capture clinically meaningful heterogeneity in a disease group may differ. In our cohort, the error was as high as 5–10 degrees in some pelvic and trunk ROM measurements. As demonstrated above, this magnitude of error may be clinically significant in some disease groups and may lead to clinically insufficient IMU accuracy. However, an IMU may have value in a clinic setting where mocap is not readily available, for monitoring disease progression over time in select cases.

In this study, we assess a normal cohort of young healthy individuals to evaluate the accuracy of IMU and its associated Mobility Lab software Version 4. Our goal was to specifically assess the performance of the Mobility Lab software for measuring the kinematics of the trunk and pelvis; therefore, we started with the first step, which is assessing the performance in a healthy cohort. Here, we present a protocol for validation. Future work should include an assessment of validity and reliability prior to clinical application in any specific patient population or environment because intrinsic (patient- or disease-related) and extrinsic (environmental; home versus clinic) factors can affect the accuracy and validity of an IMU.

The strength of this study is that the IMU technology was assessed in the same manner as it is most likely to be used in clinical practice. In a busy clinical practice, a clinician is most likely to employ an available out-of-the box software solution and may not be able to develop and validate a custom IMU configuration and analytical program that would be carried out in a biomechanical gait lab.

There are some limitations. To allow concurrent mocap, the sternum and sacrum IMU were placed directly on the skin with double-sided tape. In clinical use, the sensor would be placed on top of clothing with the provided Velcro adjustable band. The accuracy of such a real-world application compared to the current study may be lower. The trunk model assumes that the trunk is a single rigid segment, which only partially reflects real life, multi-level spinal mobility. However, this was deemed sufficient to compare the IMU (which provides a single trunk ROM value) to the mocap system. The 6-sensor IMU system used in this study is not able to capture very detailed ROM along the entire spine. Brouwer et al. used multiple spinal IMUs and modified analysis algorithms and demonstrated only a 1.1 degree error in ROM compared to the gold-standard [41]; therefore, motion analysis methods should be selected in light of the clinical or research question and commercially available IMU set-ups and software suites may not be best suited for detailed multisegmented spinal ROM analysis.

In summary, IMUs offer an inexpensive and easy solution for motion analysis. Factors affecting the measurements include sensor type, location calibration, analysis algorithm, filtering of data, tasks examined, velocity of movement, and reference segment definition [59,60]. In normal individuals in a laboratory setting, trunk and pelvis ROM values acquired using an IMU in all three planes of movement are largely accurate and reliable, with greater errors seen in the pelvis ROM, especially rotation. However, it is recommended that the error in ROM measurements be evaluated for specific patient populations before making any clinical interpretations of the data, to account for disease factors affecting measurements. IMUs should also be validated in real-world environments where the margin of error may be larger.

## Figures and Tables

**Figure 1 bioengineering-11-00659-f001:**
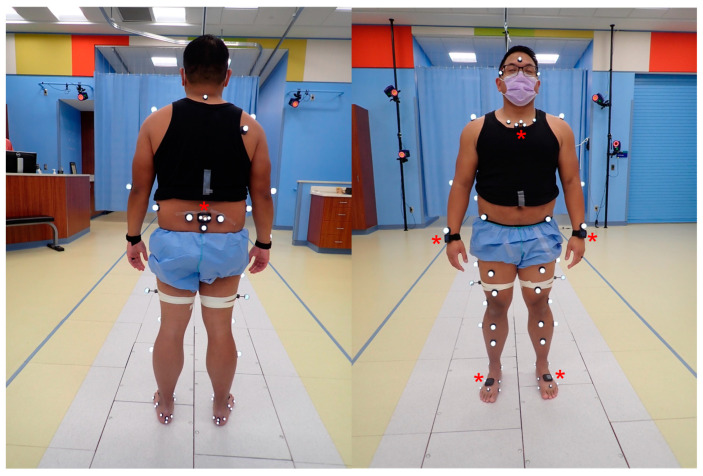
Retroreflective markers for mocap, as per the modified Helen Hayes system, and IMUs (marked by red asterisk) placed on the dorsum of the foot, wrists, sternum, and sacrum.

**Figure 2 bioengineering-11-00659-f002:**
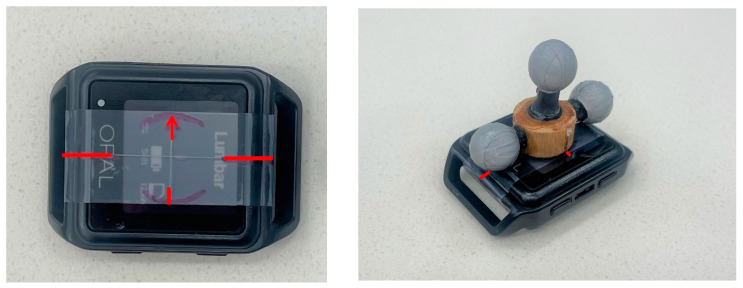
Triad marker placement on an IMU. The triad base is placed within the purple circle, and the pair of horizontal triad markers align with the long, horizontal lines (**Left Panel**). The arrow signifies the UP direction, as placed on the participant (**Right Panel**).

**Figure 3 bioengineering-11-00659-f003:**
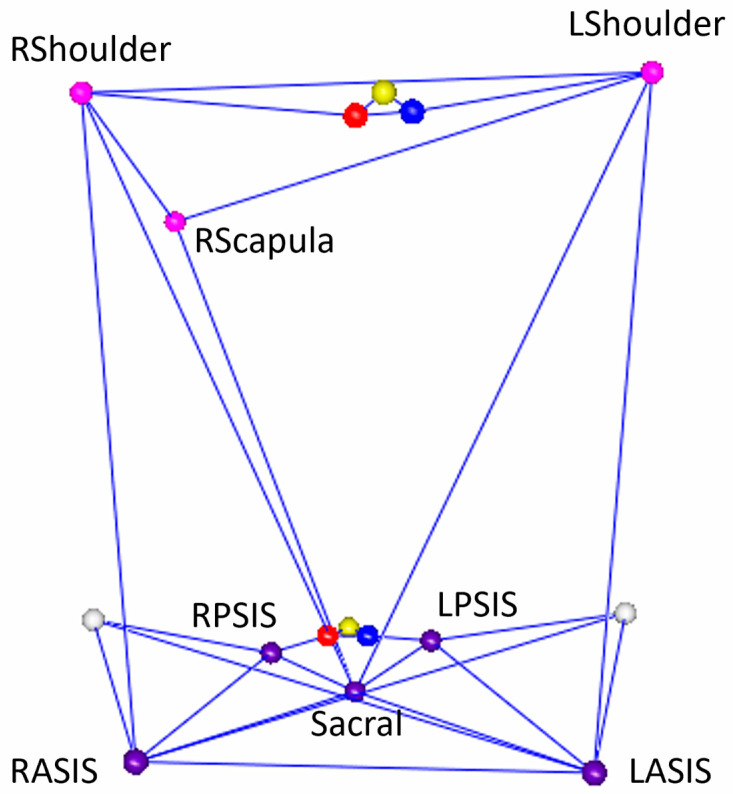
Visual depiction of the anatomical trunk and pelvis model used for mocap.

**Figure 4 bioengineering-11-00659-f004:**
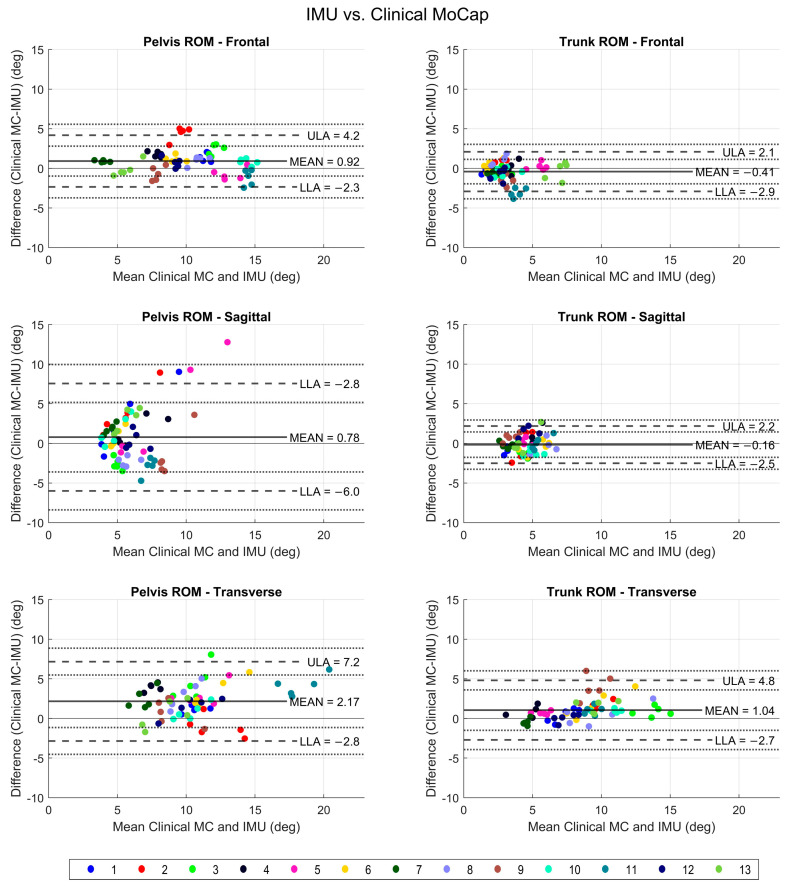
Comparison of IMU and clinical mocap ROM values for pelvis and trunk. Individual subject trials are identified by specific color for each participant. The *x* axis shows mean value; *y*-axis shows mean difference between mocap and IMU. MC: mocap, IMU: inertial measurement unit, ULA: upper limit of agreement, LLA: lower limit of agreement, SE: standard error.

**Figure 5 bioengineering-11-00659-f005:**
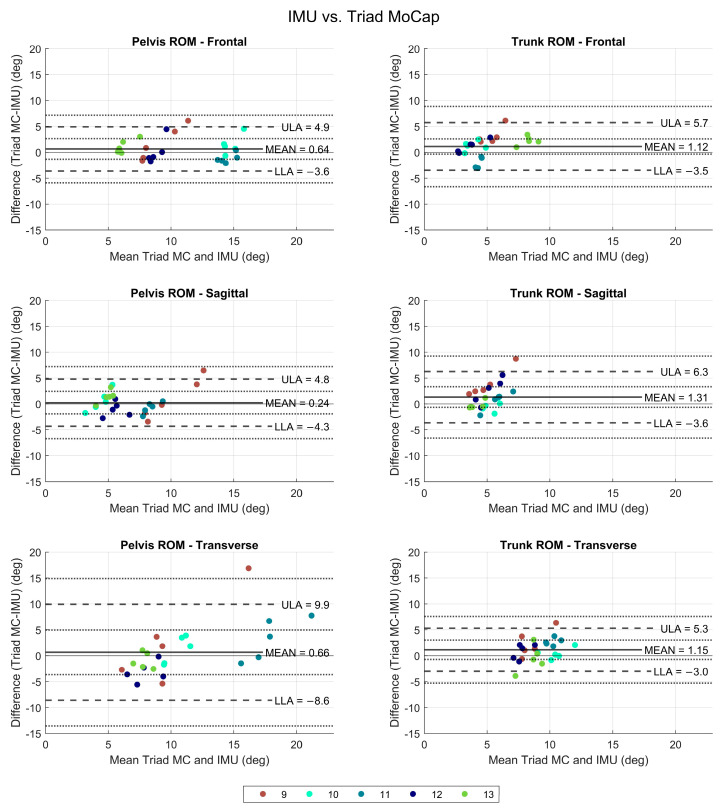
Comparison of IMU and triad mocap on a subset of five participants. Each participant’s trial is identified by the same color. *x*-axis shows mean values; *y*-axis shows difference between triad mocap and IMU. MC: mocap, IMU: inertial measurement unit, ULA: upper limit of agreement, LLA: lower limit of agreement, SE: standard error.

**Table 1 bioengineering-11-00659-t001:** Range of motion measured in degrees is shown for all three modalities. Range = Min–Max, Q1: first quartile, Q3: third quartile.

		IMU		Mocap		Triad	
		Range	Median (QI–Q3)	Range	Median (QI–Q3)	Range	Median (QI–Q3)
Pelvis	Frontal	2.8–15.8	9.3 (7.3–11.4)	3.8–15.5	11.3 (8.7–12.6)	5.8–18.1	11.9 (7.4–15.6)
	Sagittal	3.0–10.2	5.1 (4.0–6.8)	3.2–19.4	5.7 (4.5–7.1)	2.3–15.8	6.2 (5.2–7.8)
	Transverse	5.0–17.3	9.1 (7.7–10.5)	6.2–23.5	11.3 (9.9–12.7)	4.5–25.1	8.8 (6.7–14.0)
Trunk	Frontal	1.3–8.1	2.9 (2.3–4.4)	0.9–7.7	25.5 (1.9–3.6)	2.6–10.1	5.3 (3.6–7.5)
	Sagittal	2.4–7.1	4.6 (3.9–5.3)	2.2–7.2	4.2 (3.6–5.2)	3.3–11.6	5.4 (4.3–6.7)
	Transverse	2.8–14.7	8.1 (6.2–9.3)	3.3–15.3	9.4 (6.5–11.5)	5.3–13.7	9.6 (8.5–10.9)

**Table 2 bioengineering-11-00659-t002:** Reliability of IMU and mocap ROM measurements. IMU: inertial measurement unit, RMSE: root mean square error.

	RMSE Range	
	Mocap	IMU
Pelvis Frontal ROM	0.34–1.26	0.26–0.88
Pelvis Sagittal ROM	0.27–6.65	0.08–0.98
Pelvis Transverse ROM	0.64–2.61	0.26–2.27
Trunk Frontal ROM	0.37–1.04	0.26–0.75
Trunk Sagittal ROM	0.34–1.49	0.29–0.78
Trunk Transverse ROM	0.37–3.13	0.17–2.13

## Data Availability

Data will be provided upon reasonable request to the corresponding author.

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
