# Peer review of "Validation of Pelvis and Trunk Range of Motion as Assessed Using Inertial Measurement Units"

_bioengineering, 2024, doi:10.3390/bioengineering11070659_

Round 1
Reviewer 1 Report
Comments and Suggestions for Authors
Reviewer comments:
This manuscript entitled “Validation of Pelvis and Trunk Range of Motion as assessed by 2 Inertial Measurement Units” investigates assess the accuracy (validity) and reliability (consistency) of the trunk and pelvis ROM during steady-state gait in all three planes as measured by Opal APDM 6 sensor IMU system and calculated using the Mobility Lab version 4 software compared to a gold standard optoelectric motion capture system. At the same time, as the author mentioned in the limitation part, there are big problems with the current research application which is also the concern of the reviewer. Specific comments are shown below:
Abstract
1. Lines 8-9: The background section provided by the author seems insufficient, not enough to support the eliminating topic, please expand.
2. Lines 13-15: Clearly state the hypothesis that your study is designed to test. For example, if you hypothesize that the IMU provides a valid and reliable measurement for trunk 22 and pelvis ROM. Then state this clearly.
Introduction
3. Lines 30-31: This change can be due 30 to musculoskeletal or neurological disorders, which together form the largest burden of dis- 31 ability-adjusted life years lost. The reasons for this situation are suggested to be explained a little more.
4. Lines 39-40: “IMU systems have been validated in a variety of settings.” Please describe it from various environmental perspectives
5. Based on prior research, it appears that the author has not formulated reasonable hypotheses for the experiments. Please provide supplementary information in this regard.
Materials and Methods
6. Line 68: Please explain the subject carefully and kindly include the criteria for participant inclusion and exclusion in the study.
7. Line 75-76: What is the basis for wearing it in these places?
8. Lines 83-84: The reviewer suggested that the author clearly describe the specific location of the reflective markers, so that the following readers can directly refer.
9. Lines 105-106: What is the basis for this?
Discussion
10. Line 282-284: As the author mentioned in the limitation part, there are big problems in the current research method, which is also the concern of the reviewer.
11. The reviewer considers the author to be making recommendations for future research to delve into specific aspects of the Inertial Measurement Unit (IMU), or to conduct related research in the broader field of sports medicine.
Comments on the Quality of English Languageno
Reviewer 2 Report
Comments and Suggestions for Authors
The present study by F. Ali et al focuses on assessing the accuracy of the trunk and pelvis ROM during steady-state gait as measured by Opal APDM 6 sensor IMU system. The article is well-written, and the authors have articulated their message adequately. While their experimental design and methodology are comprehensive, the reviewer fails to realize the novelty of the study. There are multiple research reports which report high accuracy and reliability with IMU. However, the aim of the study could be of high interest to the readers and researchers in this field. Therefore, the reviewer thinks that the manuscript can still be considered for publication after addressing the following major and minor revision suggestions. The biggest concern for the reviewer is the novelty of the study.
· Is there a reason why participants of only certain age groups were chosen? In the real world, clinicians will most likely use these devices on perhaps older or injured individuals. The reviewer thinks that it would add immense value to the manuscript if the data for different age groups/types is included in the manuscript. Furthermore, the findings related to IMUs were already reported elsewhere, therefore, it is even more important to include more data on different age groups.
· The reviewer suggests authors to consider editing the abstract to reflect more on the importance of this study. At present, the abstract depicts the aim, results, and conclusions of the study. It might be more useful for the readers to know about the outlook of the study, and how it can benefit the clinicians and researchers.
· Editing of figures and their captions can be improved – figures are not aligned well with the manuscript text and the text for figure captions.
· Figures 4 and 5 can be arranged in portrait mode rather than landscape mode to be consistent with the manuscript style.
· On multiple instances the style of referencing is inconsistent. Usually, the reference is mentioned at the end of a sentence (before full stop) and not at the beginning of the sentence.
Comments on the Quality of English LanguageThe quality of English in manuscript is adequate
Author Response
Response to Reviewer Comments are attached.

Round 2
Reviewer 2 Report
Comments and Suggestions for Authors
The reviewer would like to thank the authors for their detailed response to the concerns raised. After reviewing their manuscript, the reviewer is satisfied with the changes made in the manuscript. Overall, I am satisfied with the manuscript and recommend its publication.
Comments on the Quality of English LanguageThe quality of English in the manuscript is at par with an average English reader and apart from a few minor errors, the message of the manuscript is explained adequately.
